# Y-TZP Physicochemical Properties Conditioned with ZrO_2_ and SiO_2_ Nanofilms and Bond Strength to Dual Resin Cement

**DOI:** 10.3390/ma15227905

**Published:** 2022-11-09

**Authors:** Ricardo Faria Ribeiro, Danilo Flamini Oliveira, Camila Bussola Tovani, Ana Paula Ramos, Ana Flavia Sanches Borges, Adriana Claudia Lapria Faria, Rossana Pereira de Almeida, Renata Cristina Silveira Rodrigues

**Affiliations:** 1Department of Dental Materials and Prosthodontics, Ribeirao Preto School of Dentistry, University of Sao Paulo-FORP-USP, Ribeirao Preto 14040-904, SP, Brazil; 2Department of Chemistry, Faculty of Philosophy, Sciences and Letters at Ribeirao Preto, University of Sao Paulo-FFCLRP-USP, Ribeirao Preto 14040-901, SP, Brazil; 3Department of Operative Dentistry, Endodontics and Dental Materials, Bauru School of Dentistry, University of Sao Paulo-FOB/USP, Bauru 17012-901, SP, Brazil

**Keywords:** Y-TZP zirconia, bond strength, surface treatment

## Abstract

Commercial Yttria-tetragonal zirconia polycrystalline (Y-TZP) was subjected to surface treatments, and the bond strength of dual resin cement to Y-TZP and failure modes were evaluated. Disks (12 mm × 2 mm), cylinders (7 mm × 3.3 mm), and bars (25 mm × 5 mm × 2 mm) were milled from Y-TZP CAD-CAM blocks, divided into seven groups, and subjected to different surface treatments; silicatization was used as control. On the basis of the literature, this study evaluated modifications with films containing SiO_2_ nanoparticles and silane; SiO_2_+ZrO_2_—SiO_2_ (50%) and ZrO_2_ (50%) nanoparticles, SiO_2_+ZrO_2/_Silane-SiO_2_ (50%) and ZrO_2_ (50%) nanoparticles, and silane. Specimens were analyzed by wettability (n = 3), surface free energy (n = 3), X-ray diffraction (n = 1), Fourier transform infrared spectroscopy (FTIR) (n = 1), roughness (n = 5), shear bond test (n = 10), and dynamic modulus (n = 3). Specimens treated with hydrofluoric acid—HF 40% presented significantly higher contact angle and lowest surface free energy (*p* < 0.05). The SiO_2_/Silane presented crystalline SiO_2_ on the surface. The surface roughness was significantly higher for groups treated with nanofilms (*p* < 0.05). Shear bond strength was significantly higher for silicatization, HF 40%/silicatization, SiO_2_/Silane, and SiO_2_+ZrO_2_/Silane groups. The proposed treatments with nanofilms had potentially good results without prejudice to the physicochemical characteristics of zirconia. Generally, groups that underwent silica surface deposition and silanization had better bond strength (*p* < 0.005).

## 1. Introduction

Dentists and patients prefer restorations that simulate natural teeth. Ceramics have been developed to enable metal-free restorations with excellent aesthetics and high mechanical performance [1,2,3,4,5,6,7,8]. Yttria-tetragonal zirconia polycrystalline (Y-TZP) has been used in both conventional and implant prostheses because of its excellent mechanical properties and biocompatibility, being easily milled in computer-aided design/computer-aided manufacturing (CAD-CAM) equipment [3,7,9,10,11,12,13].

The clinical success of these restorations depends on a stable and robust bond with the dental tissues [8,13,14,15,16,17,18]. Y-TZP still has problems with adhesion stability because of its microstructural characteristics, lack of silica, and glass matrix that comprise the etching and silanization process [8,12,19,20].

Factors such as wettability, surface free energy, surface roughness, and chemical adhesion can directly influence bond strength [21,22,23].

Airborne particle abrasion and chemical surface treatment schemes have been proposed to alter the surface properties of Y-TZP and improve the adhesion to the underlying structure [8]. Airborne particle abrasion with alumina particles (50 to 125 µm) was proposed by promoting surface irregularities and increasing the wettability through bond agents [8,24,25,26,27,28,29]. However, some authors argued that airborne particle abrasion cause microcracks on the Y-TZP surface, leading to premature catastrophic failure [26], while others claim that these treatments inhibit crack propagation, increasing the resistance [14,30,31].

Acids were also used at different concentrations and times to favor adhesion [32,33,34,35,36]. In addition, acid etching at elevated temperatures (70–80 °C) has been tested, showing higher bond strength by increasing molecular activity of etching in a short time, promoting sharp and rougher surface than acids used at room (20–25 °C) temperature [37]. Another study that evaluated the effect of HF etching in different concentrations and temperatures showed that the increase in temperature and HF concentration increased the surface roughness of Y-TZP; however, shear bond strength was similar for HF 40% at room temperature, HF 20%, and HF 40% at 70–80 °C [38]. Similarly, acid etching by inducing a phase transition (decreasing the performance of Y-TZP) is also controversial [32].

Resin primers or cement containing 10 methacryloloyloxydecyl dihydrogen phosphate (10-MDP) monomers have also presented conflicting results. Several studies have shown improved bonding strength to Y-TZP because the acid contains a chemical structure with long, hydrophobic spaced carbon chains that can form double bonds with zirconia [16,20,34,39,40,41,42,43,44]. In contrast, other studies have shown that the chemical bond between Y-TZP and MDP molecules is prone to hydrolysis at high humidity, impairing the bond [16].

Tribochemical silica coating is a treatment that combines mechanical surface abrasion (Al_2_O_3_) with chemical functionalization by silanization of silica particles. Therefore, Y-TZP presents silica in its surface layer, allowing the compatibility between inorganic and organic surfaces [26,28,39,40,41,42,45,46,47,48,49]. The deposited coating layer is chemically bonded to zirconia and has also been reported to be reactive with organosilane coupling agents [50].

The effect of nonthermal atmospheric pressure plasma treatment was also proposed by chemically modifying the surface of Y-TZP without compromising mechanical properties. A previous study showed that surface energy increased after this treatment, but roughness was not affected; Thus shear bond strength of this treatment was better than control (without any treatment) but lower than sandblasting [51]. The association of this treatment with primers did not improve the shear strength of resin to Y-TZP [52]. Sonochemical treatment before and after sintering was also compared to sandblasting to alter the surface of Y-TZP and affect shear bond strength. Although sonochemical treatment alters roughness in a nanoscale while sandblasting alters in a microscale, both treatments presented similar shear bond strength [53].

Resin types of cement have an adequate bond strength to ceramics because they have satisfactory flow, wettability, and chemical interaction. Thus, changes in topography and surface modify the total area and its physicochemical features, improving the adhesive system wetting [27,54,55,56,57].

Several surface treatments have been proposed to modify surface features in order to improve shear bond strength by interfering with the mechanical and chemical interaction with Y-TZP, but any substantial improvement has not been noted yet. Thus, the present study aimed to modify the surface of Y-TZP using ZrO_2_ or SiO_2_ nanofilms in the search for better chemical interaction with 10-MDP of resin types of cement or silane, respectively. 

The objective of this study was to characterize the surface of Y-TZP subjected to different surface treatments and to evaluate the bond strength and failure mode of Y-TZP obtained by CAD-CAM. The aim of this study is to investigate the potential of different nanofilms used to treat the zirconia surface, comparing them to other treatment methods already used, characterizing the treated surfaces, and testing the bond strength. The research hypothesis is that the different surface treatments do not alter the Y-TZP surface when compared with silicatization (control), not improving the bond strength of dual resin cement to Y-TZP.

## 2. Materials and Methods

This study was divided into Part 1—Surface treatments and shear bond strength and Part 2–Dynamic modulus.

The number of specimens was based on previous studies, as cited for each test. The ideal number of layers to be deposited on the surfaces was determined in a pilot study according to the changes in the contact angle (increase in wettability), surface free energy (increase in free energy), and morphological analysis of the surface in SEM.

### 2.1. Part 1—Surface Treatments and Shear Bond Strength 

Commercial Y-TZP (In Coris ZI; Sirona Dental; Composition: Zirconium dioxide + hafnium dioxide + yttrium trioxide >99 wt.%, aluminum trioxide <0.5 wt.%, and other oxides <0.5 wt.%) disks and cylinders were milled (Cerec inLab MCXL; Sirona Dental) from CAD-CAM blocks following manufacturer instructions, and after sintered (inFire HTC Speed; Sirona Dental) using a specific sintering cycle provided by Sirona Dental achieving the desired dimensions (12 mm × 2 mm for disks, and 7 mm × 3.3 mm for cylinders). Disks were divided into 7 groups according to surface treatment, as shown in Figure 1:

Silicatization: airborne-particle abrasion with 110 μm silica-modified Al_2_O_3_ (Rocatec Plus; 3M ESPE) for 15 s under 3 bar pressure and distance of 20 mm, with the application of a silane coupling agent [RelyX Ceramic Primer, 3M ESPE. Composition: 3-MPS-methacryloxypropyltrimethoxysilane (pre-hydrolyzed silane), ethanol, water [49]–control group;

HF—Treatment with Hydrofluoric Acid (HF) 40% for 210 s and washing with water for 30 s [34];

HF/silicatization—treatment with HF 40%, and after silicatization and application of a silane coupling agent (RelyX Ceramic Primer);

ZrO_2_—Surface modification with a film containing ZrO_2_ nanoparticles obtained by the sol–gel process;

SiO_2_/Silane—surface modification with a film containing SiO_2_ nanoparticles, obtained by the sol–gel process and application of a silane coupling agent (RelyX Ceramic Primer);

SiO_2_+ZrO_2_—surface modification with a film containing SiO_2_ (50%) and ZrO_2_ (50%) nanoparticles obtained by the sol–gel process;

SiO_2_+ZrO_2_/Silane—surface modification with a film containing SiO_2_ (50%) and ZrO_2_ (50%) nanoparticles, obtained by the sol–gel process and application of a silane coupling agent (RelyX Ceramic Primer).

Cylinders were not submitted to any surface treatment simulating the interior surface of a monolithic zirconia crown cemented over a zirconia abutment represented by the disk surface.

Colloidal dispersions containing ZrO_2_ nanoparticles were prepared by the hydrolysis of the precursor oxide-zirconium chloride (ZrOCl_2_.8H_2_O-Sigma/Aldrich 99%). The salt was initially dissolved in ethanol and heated at 60 °C for 1 h. The average size of the particles obtained was 50 nm.

Colloidal dispersions of SiO_2_ particles were prepared from the precursor ethyl tetraethyl orthosilicate solution (TEOS-Si(CH_2_CH_3_O)_4_/Sigma-Aldrich 99%) containing water and hydrochloric acid as a catalyst for hydrolysis.

To prepare the mixed colloidal dispersions of ZrO_2_:SiO_2_ with a Zr:Si 1:1 molar ratio, the dispersion of ZrO_2_ nanoparticles in ethanol was mixed with the solution of tetraethyl orthosilicate. The TEOS:HCl:ethanol:H_2_O ratio was 1:0.01:37.9:2. The mixture was maintained for 1 h at 65 °C and was ready to be used in preparing the coatings after 16 h at room temperature. The size of the mixed particles, determined by dynamic light scattering, was 100 nm.

A preliminary study was performed regarding the ideal number of layers to be deposited on the surfaces. Films were prepared to contain 1, 5, 10, and 20 layers of each precursor solution. The contact angle (wettability), surface energy, and surface qualitative analysis were performed by scanning electron microscopy on the specimen. For each number of pre-established layers and nanofilm proposed, 3 disks were obtained for a total of 36 disks to assess wettability and surface energy. One disk was preserved without any treatment for comparison.

The Y-TZP disks were covered with nanoparticles from the prepared dispersions with the aid of a dip-coater system. For this, they were immersed for 5 min and calcinated at 900 °C for 1 min, and for deposition of the subsequent layers, each disk was immersed again in the respective dispersions for 5 min, calcinated for 1 min, and cooled for 5 min.

The definition of the number of layers to be deposited on the disc surfaces for groups ZrO_2_, SiO_2_/Silane, SiO_2_+ZrO_2,_ and SiO_2_+ZrO_2_/Silane was developed from a pilot study, in which wettability characteristics, surface energy and surface morphology of the samples were evaluated. It was possible to define an ideal for the subsequent tests to modify the surfaces of the ZrO_2_ group with 10 layers of nanoparticles since this number of layers promoted the lowest values of contact angle and the highest values of surface energy. For the SiO_2_+Silane group, the amount of 5 layers was defined. In this group, there were no differences between the amounts of layers when analyzing the contact angle. Despite the lower surface energy found for 5 layers, this amount was selected considering the high absolute value obtained. Considering the similar wettability between the groups, working with the application of only one layer of silica oxide nanoparticles could generate little deposition of surface silica, which is a fundamental part of adhesion. Not using more than 5 coats would be justified because there is no difference between wettability, and applying 10 or 20 coats would take a long time and increase costs. In the SiO_2_+ZrO_2_ group, 5 layers were selected considering the lowest wettability found (similar to 10 layers) and the statistical similarity for surface energy between the layers. Likewise, the application of 10 layers would be time and cost-intensive without considerable improvements in the evaluated characteristics.

Silicatization and HF/silicatization groups were initially air-abraded with 110 µm alumina particles under 3 bar pressure for 15 s, 20 mm from the Y-TZP surface, and after being subjected to silicatization (110 µm silica-coated alumina particles (SiO_2_; Rocatec Plus; 3M ESPE) [47]. A silane coupling agent (Rely X Ceramic Primer; 3M ESPE) was applied in silicatization, HF/silicatization, SiO_2_, and SiO_2_+ZrO_2_/Silane groups for 60 s and dried to achieve chemical bonding.

#### 2.1.1. Wettability and Surface Free Energy

Disks (n = 3) were used to measure the contact angles, wettability, and surface free energy, using a goniometer (OCA-20 Dataphysics (Stuttgart, Germany); Dataphysics Instruments) [23,58,59]. The sessile drop technique with three liquids (water, diiodomethane, and formamide) with different surface tensions was used. The image of a drop on the surface was registered. Three measurements of the contact angle on each specimen were performed, and the average was determined. The values of contact angles, surface tension (mJ.m^−2^), and dispersive and polar components (mJ.m^−2^) were calculated using software based on the formula of Owens–Wendt–Kaelble [60].

#### 2.1.2. X-ray Diffraction (XRD)

To determine the alterations in the crystalline phase of the disks resulting from the surface treatments, XRD patterns from one specimen for each group were obtained using a diffractometer (Bruker-ASX D 5005; Bruker Corp (Karlsruhe, Germany)) with Ni-filtered Cu-Kα radiation. The diffraction peaks were indexed on the basis of the power diffraction files (PDF) of the International Centre for Diffraction Data (ICDD, Newtown Square, PA, USA) [61].

#### 2.1.3. Fourier Transform Infrared-Attenuated Total Reflectance (FTIR-ATR)

Spectra were obtained from the same XRD specimens by FTIR spectroscopy (Shimadzu IR-Prestige 21; Shimadzu Corp., Kyoto, Japan) with an attenuated total reflectance (ATR) sampling accessory. This technique is used for the qualitative and semi-quantitative characterization of a specimen.

#### 2.1.4. Roughness

The surface roughness of the disks (n = 5) was determined using a 3D laser confocal microscope (LEXT 3D Measuring Laser Microscope OLS4000; Olympus) [58,62]. Objective lenses of 100× and 2132× magnification were used. Measurements were performed with specimens presintered, sintered, and after the treatments. Magnified images (108 and 432×) were digitally analyzed to calculate the surface roughness Sa as the arithmetic average of the 3D roughness.

#### 2.1.5. Shear Bond Test

Y-TZP (In Coris ZI; Sirona Dental Systems GmbH, Bensheim, Germany) CAD-CAM cylinders (n = 10) [63,64,65] were cemented using dual-cure resin cement (Panavia F 2.0; Kuraray, Tokyo, Japan), onto the Y-TZP disks (n = 5) that were embedded in acrylic resin with the treated surface exposed. Two cylinders were individually cemented on each disk. Cementation procedures were performed using a parallelometer to ensure that the cylinder was positioned perpendicular to the disk surface, and a uniform layer of resin cement was ensured by using a 200 g weight throughout the process. Cement was light-cured for 80 s, 20 s on each side (Emitter C, Schuster Equipamentos Odontológicos–1250 mW/cm^2^). Oxygen-blocking gel (Oxyguard II; Kuraray medical Inc., Tokyo, Japan) was applied at the margins after the light-polymerizing procedure, and the cement was light-polymerized again for 80 s as described above. Specimens were stored at 100% humidity at 37 °C for 24 h. In this study, no artificial aging was performed to avoid interference in the cementation process. The idea was to characterize the potential best outcome by including the subsequent aging study based on the results obtained now. The shear bond strength test was performed according to ISO/TR 11405 [66] in a universal testing machine (BioPdi, São Carlos, SP, Brazil) using a chisel knife at 0.5 mm/min speed and load cell of 100 N. The shear bond strength (MPa) was calculated considering the maximum force (N) and bond area (mm^2^) based on the diameter of each cylinder. After testing with the first cylinder, a second cylinder was luted on the other areas of the Y-TZP disk and subjected to shear tests under the same conditions described. Each cylinder was cemented and tested individually.

All specimens were analyzed under a stereomicroscope (S8APO; Leica Microsystems, Wetzlar, Germany) at 40× magnification. The failure mode was classified as adhesive (between dual-cure resin cement and surface treated of the disk), mixed (adhesive and cohesive failure of dual-cure resin cement). When dual-cure resin cement was noted at the surface of the treated disk, indicating failure between cement and cylinder or failure of cohesive forces of the cement, it was not possible to determine where the failure occurred, and the term unclassifiable term is used. The results are expressed as %.

### 2.2. Part 2: Dynamic Modulus

Bar-shaped specimens (n = 3) [67,68] were cut from CAD-CAM blocks (InCoris Zi Maxi L; Sirona Dental) using a precision saw. Sintering was performed according to the manufacturer’s instructions. They were analyzed in three conditions: presintered (40 mm × 6.25 mm × 2.5 mm), post-sintered (25 mm × 5 mm × 2 mm), and after surface treatments, as described previously.

The elastic modulus was evaluated by the impulse excitation technique (ASTM E-1876) using Sonelastic equipment (ATCP Engenharia Física). Bending excitation was used. [68]

#### Statistical Analysis

The surface free energy and wettability data were analyzed by nonparametric Kruskal–Wallis and Dunn tests. The surface roughness and dynamic modulus data were analyzed using a linear model of mixed effects because of intragroup (presintering, postsintering, and post-treatment) and intergroup (silicatization to SiO_2_+ZrO_2_/Silane) comparisons. Multiple comparisons were made by orthogonal contrasts using PROC GLM from SAS version 9.3 (SAS Institute Inc. (Cary, NC, USA)). The shear bond strength data were compared by one-way analysis of variance (ANOVA) and the Tukey test. To analyze whether there was an association between the failure types and the groups, a chi-square test was performed. A significance level of 5% was used.

## 3. Results

Wettability and surface free energy results are presented in Table 1. Silicatization and HF groups presented the lowest contact angles that the software was not able to measure. ZrO_2_ and SiO_2_/Silane groups presented contact angles significantly lower than HF. HF presented the highest contact angle and lowest surface energy. The use of silane in silicatization, HF/silicatization, and SiO_2_/Silane groups was important to increase wettability (low contact angles) and surface free energy. With the SiO_2_+ZrO_2_, after the silane application, the drop spread completely. The silane increased wettability and caused the drop to spread in such a way that the equipment was not able to measure the angle between the drop and the disk. So, the silane was responsible for increasing wettability, consequently increasing the surface energy. The application of silane would be necessary because of the presence of silica on the surface of the disk, but it could be omitted depending on what we found in relation to the other groups. After this, there were no results for the SiO_2_+ZrO_2_ /Silane group.

All groups presented similar XRD patterns except for SiO_2_/Silane group (Figure 2). In this group, a peak at 2θ = 22° (red *) indicated the presence of crystalline SiO_2_ (cristobalite PDF 39–1425) at the surface.

Spectra generated from the surface of the treated disks are presented in Figure 3.

The deposition of SiO_2_ (in the SiO_2_/Silane group) was confirmed by FTIR. The spectrum shows the presence of the band at 1070 cm^−1^ and a secondary band at 800 cm^−1^, both attributed to O-SiO. It was not possible to analyze the chemical composition of the specimens in ZrO_2_ and SiO_2_+ZrO_2_ groups using this technique because the ZrO_2_ results in bands between 500–600 cm^−1^, which is outside the detector range of the reflection accessory used.

For surface roughness, there was a significant difference among groups (*p* < 0.05), among Y-TZP processing stages (*p* < 0.05), and group*Y-TZP processing stages (*p* < 0.05). As group*Y-TZP processing stages was significant, groups presented different behavior at each Y-TZP processing stage. Before and after sintering, surface roughness was statistically similar once any surface treatment was performed, but after treatments, groups treated with nanofilms (ZrO_2_, SiO_2_/Silane, SiO_2_+ZrO_2_, SiO_2_+ZrO_2_/Silane groups) presented higher surface roughness than control and groups treated with HF (Table 2).

Representative images of each group submitted to 3D confocal microscopy are presented in Figure 4. There were more evident changes in the surface morphology of ZrO_2_, SiO_2_/Silane, SiO_2_+ZrO_2_, and SiO_2_+ZrO_2_/Silane groups. In silicatization, HF, and HF/silicatization groups, the surface treatments promoted uniform modifications.

The shear bond strength of groups HF/silicatization, SiO_2_/Silane, and SiO_2_+ZrO_2_/Silane was statistically similar to silicatization and higher than other groups, especially ZrO_2_ (*p* = 0.005), which displayed lower values of shear bond strength, but without a significant difference to groups HF and SiO_2_+ZrO_2_ (Table 3).

To analyze whether there was an association between the types of failures and the groups, a chi-square test was performed, in which the likelihood ratio showed no association between these variables (*p* = 0.569) (Figure 5 and Figure 6).

No significant differences were found among groups in the dynamic modulus (*p* = 0.605). All groups presented similar behavior at each processing stage. Dynamic modulus at presintering was statistically lower than postsintering and post-treatment stages (*p* < 0.05) (Table 4).

## 4. Discussion

This study aimed to investigate the effects of nanofilms used to treat Y-TZP surface on the surface roughness, wettability, surface free energy, surface morphology, dynamic modulus, and bond strength of Y-TZP to dual resin cement. The hypothesis that different treatments would not affect the Y-TZP adhesive properties was rejected.

Y-TZP has little or no vitreous portion, and the lack of silica makes the adhesion to resinous materials difficult. Thus, for effective adhesion, specific surface treatments to favor the bonding of Y-TZP to resin cement are required [20,27,42,50].

Silicatization and HF/silicatization decreased contact angles and improved wettability, probably due to the formation of a homogeneous layer, as can be visualized on laser confocal microscope images (Figure 4). Roughness was not increased, with only modification of the superficial morphology. Air abrasion with 110 μm alumina particles previously to the silicatization probably results in greater surface modification than HF etching at the proposed concentration and time, explaining the surface roughness (Sa) decrease found in HF/silicatization and the increase in surface free energy (Table 2).

Etching with 40% HF significantly reduced the wettability and surface free energy, possibly due to the presence of new chemical groups originating from the chemical reaction of HF and ZrO_2_ (Table 1). This group exhibited a complex band in the region of ~1000 cm^−1^, related to the link formation of O-F and O-Zr-F present in the structure of the oxy-fluorides of Zr (Figure 3) when analyzed by ATR-FTIR. This band indicates that there was a chemical reaction between the acid and Y-TZP, interfering with the surface free energy. Silicatization partially removes this deposited material, which can be evidenced by band distortion (Figure 3), increasing wettability and surface free energy. The deposition of approximately 40 nm nanoparticles on the HF-treated surface results in the formation of a new Zr-oxyfluoride phase, as identified by ATR-FTIR.

Increased roughness (Table 2) also has the potential to raise surface free energy [22,56], as observed in ZrO_2_, SiO_2_/Silane, and SiO_2_+ZrO_2_/Silane groups (Table 1). The groups treated by nanofilm deposition presented the highest surface roughness and surface energy. These treatments caused irregular surfaces, as observed by confocal laser microscopy (Figure 4), which was responsible for the surface roughness increase. Silicatization and HF/silicatization groups, despite having modified surfaces, showed a uniform modification pattern (Figure 4) with lower surface roughness than ZrO_2_, SiO_2_/Silane, SiO_2_+ZrO_2_, and SiO_2_+ZrO_2_/Silane groups. The surface roughness of these groups was lower than the others and statistically similar to the post-sintering disks, except in HF/silicatization (Table 2). The surface roughness of the untreated disks is caused by the milling cutter action during the CAD-CAM process because the disks did not receive any surface polishing.

Comparing the pre- and postsintering groups (Table 2), it is noted that the surface roughness of all groups was statistically similar at both stages. However, sintering increased the roughness of all groups. The Y-TZP sintering can lead to shrinkage with up to 25% volume reduction, promoting space closure [69] and leading to the modification of the topography and higher roughness of the specimens.

Groups silicatization, HF/silicatization, and SiO_2_/Silane had the highest polar component of surface free energy (γ_S_^p^) (Table 1) and had the highest bond strength (Table 3). These results show that the presence of silica followed by silanization is essential for good adhesion [25,41,45,56], proving that the silane applied to the surface of the disk prior to fixation interacts with hydrophilic surfaces. Silanes are chemicals composed of bifunctional molecules responsible for linking the organic phase (resin cement matrix) to the inorganic filler particles (deposited surface silica), forming a Si–O–Si covalent bond with Si [48]. The resin cement links to silane, preferably through its hydrophilic polymerizing organic component, explaining the highest shear strength of the groups where the polar component of the surface free overcomes the dispersive one (nonpolar) (Table 1).

The group SiO_2_+ZrO_2_, which was treated with a nanofilm of Si and Zr, displayed statistically lower bond strength because no silanization was performed. Silicatization promoted better adhesion, corroborating studies that suggested that this method was effective for Y-TZP surface treatment [26,27,39,45]. SiO_2_+ZrO_2_/Silane group also showed shear strength values statistically similar to silicatization, HF/silicatization, and SiO_2_/Silane. In SiO_2_+ZrO_2_, the value of the polar component was reduced (Table 1), but the application of silane probably increased the surface energy and wettability, making the silica surface compatible with the organic matrix of the cement and higher bond strength. Silicatization and HF/silicatization present the lowest contact angle and higher surface energy, allowing us to infer that the spreading of both silane and resin cement on the treated surface of Y-TZP is favored.

The chemical composition seems to have a greater influence than roughness in the adhesive properties. The ZrO_2_ group, which was coated with a nanofilm of ZrO_2_, presented the lowest shear strength (4.25 ± 3.0 MPa), despite having high roughness (Table 2). This was possibly due to the absence of silica preventing the chemical bonding of the dual resin cement and the surface modified by the silane. Although the treatment provided rougher surfaces (5.2 ± 0.27 µm) than the untreated disks (3.29 ± 0.22 µm) and, although the contact angles and surface energy are statistically similar to the other groups with nanofilms (Table 1), the adhesive strength was not the same. The absence of silica on the surface probably leads the adhesive resistance at the expense of micromechanical retention. The same occurred with the HF group, which had similar surface roughness to silicatization and HF/silicatization but had the lowest shear bond values (Table 3). The acid modified the surface (Figure 4), creating micromechanical retention but not chemical bonding. The presence of ZrO_2_ may also contribute to the bond decrease, reducing the surface energy and making chemical bonding unfeasible. Silicatization after conditioning acid can remove the oxy-fluoride layer, improving the surface energy and bond strength, as verified by the results of group HF/silicatization (9.14 ± 2.41 MPa) compared to HF (7.62 ± 2.74 MPa). The similar bond strength between HF and HF/silicatization can be explained by the presence of MDP in the resin cement composition.

Analyzing whether there was an association between the failure types and the groups using the chi-square test, there was no association between these variables (*p* = 0.569). Mixed failures were due to the characteristics of the dual resin cement used, which was composed of MDP with metal oxide affinity as ZrO_2_ [20,32,35,36,50,54]. The chemical bond of the resin cement probably occurred due to the MDP action over the nanoparticles of Zr deposited or over the Y-TZP surface, forming P-O-Zr covalent bonds [48].

The adhesion of thin silicon dioxide films deposited on Y-TZP surfaces is described in the literature [57]. The deposition of this film allows for chemical bonding between the deposited silica, silane, and dual resin cement. In this study, this happened when the Y-TZP surface received SiO_2_ nanofilm layers.

Thermomechanical stress and water hydrolysis of the resin-cement organic matrix and MDP-Zr interaction appear to negatively affect the bonding to zirconia [16,56]. In the current study, after cementation, the specimens were stored in a humid environment for 24 h without submitting to artificial aging that could stress the adhesive bond prior to the shear tests, and spontaneous failures were not verified prior to the execution of the shear tests. If the specimens had been subjected to artificial aging, the shear strength and failure patterns probably could have been different from spontaneous adhesion failures.

The Y-TZP phases were evaluated by XRD. No tetragonal-monoclinic phase transformations were observed. Surface treatments were not able to generate martensitic transformations on the material surface [16].

The surface treatment did not affect the dynamic modulus of the samples. Comparing the values before sintering with that after sintering and treatment, there was a significant difference once the sintering of Y-TZP resulted in sintering shrinkage with volume reduction and void closure [49], thus changing the dynamic modulus.

In the absence of a tetragonal-monoclinic transformation verified by XRD (Figure 2), and while the dynamic modulus of the treated material remains unchanged, the different treatments do not harm the material structure.

This study has several limitations. First, the samples were not thermally cycled prior to adhesion testing. Second, the surface treatment is only performed on the surface of the disk, not on the cylinder. It is then impossible to classify cohesive failures because it is impossible to accurately identify whether the visible damage affects only the resin cement or the interface between the resin cement and the untreated surface. Therefore, these are defined as unclassified errors.

Despite silicatization being considered the gold standard for the tribochemical preparation of the Y-TZP surface for cementation and bond strength, it has been difficult to supply the silica-modified Al_2_O_3_ material on the market. Thus, numerous studies were motivated to search for alternatives to obtain better results in the cementation of zirconia pieces.

Although the results obtained by nanoparticle surface treatment are promising for some studied groups, new research should be carried out to improve nanoparticle solution dispersions, and improve their obtaining and application techniques, while considering the possibility of application and infiltration of nanoparticles on presintered zirconia. It seems feasible to use nanotechnology to coat Y-TZP and improve its adhesion. It should be noted that the technique for synthesizing nanoparticles using the sol–gel method is simple, not requiring complex and expensive equipment, presenting the potential to be fabricated on a commercial scale and available to clinicians.

## 5. Conclusions

The results obtained using the proposed nanofilms for SiO_2_+ZrO_2_/Silane; SiO_2_/Silane groups indicate good potential for obtaining high bond strength values. Generally, the groups that underwent superficial silica deposition and silanization had the best bond strength.

## Figures and Tables

**Figure 1 materials-15-07905-f001:**
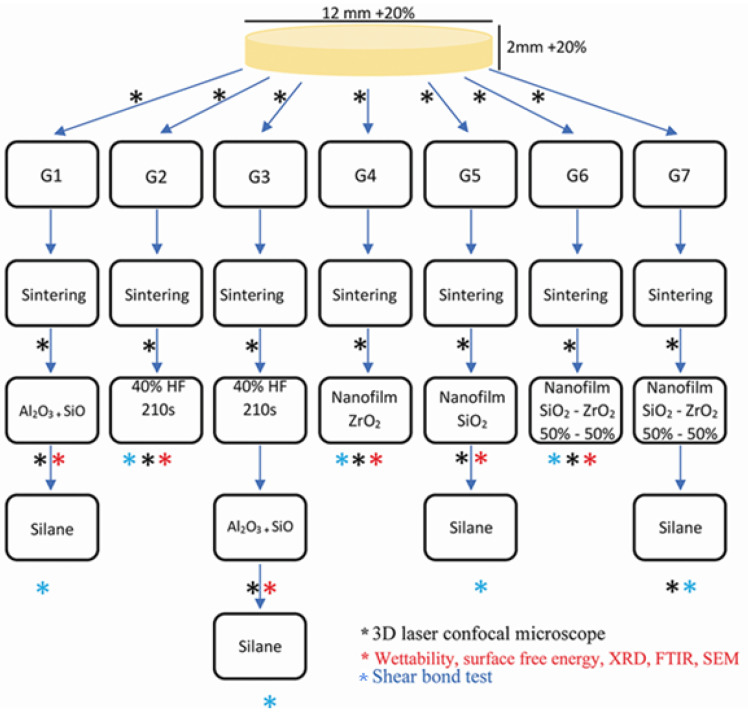
Study workflow.

**Figure 2 materials-15-07905-f002:**
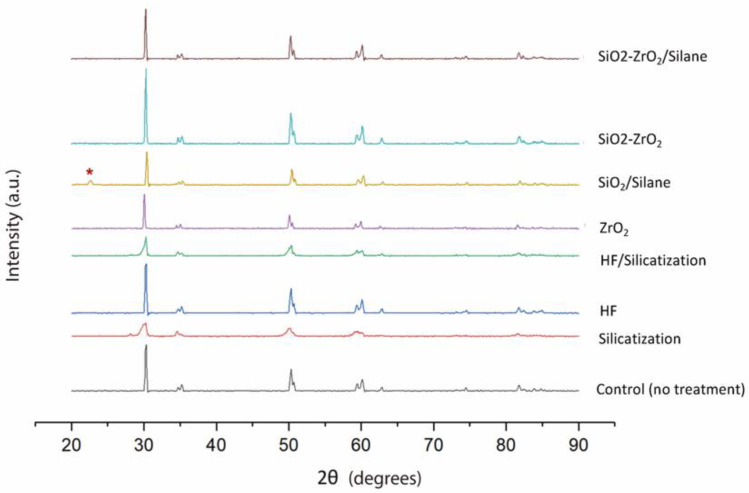
X-ray diffraction patterns for Y-TZP disks before and after different treatments. Untreated and coated disks with ZrO_2_ film have the same diffraction characteristics. Red * indicated the presence of crystalline SiO_2_ at the surface.

**Figure 3 materials-15-07905-f003:**
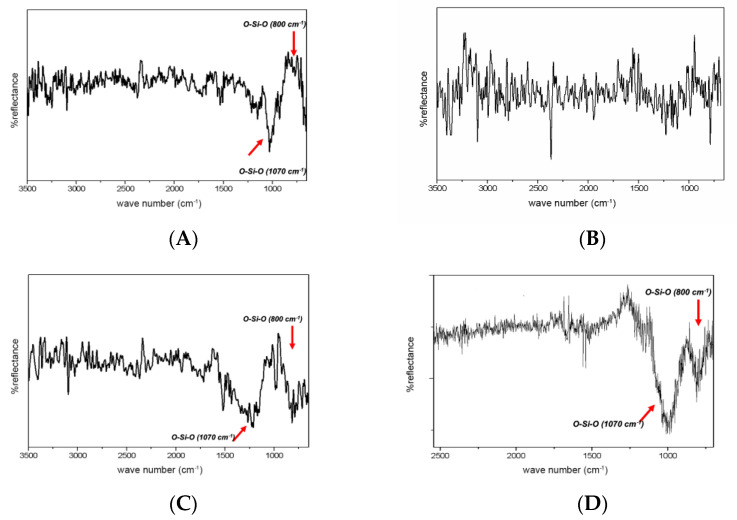
ATR-FTIR spectra of specimens after surface treatments: (**A**) silicatization; (**B**) HF; (**C**) HF/silicatization; (**D**) SiO_2_/Silane.

**Figure 4 materials-15-07905-f004:**
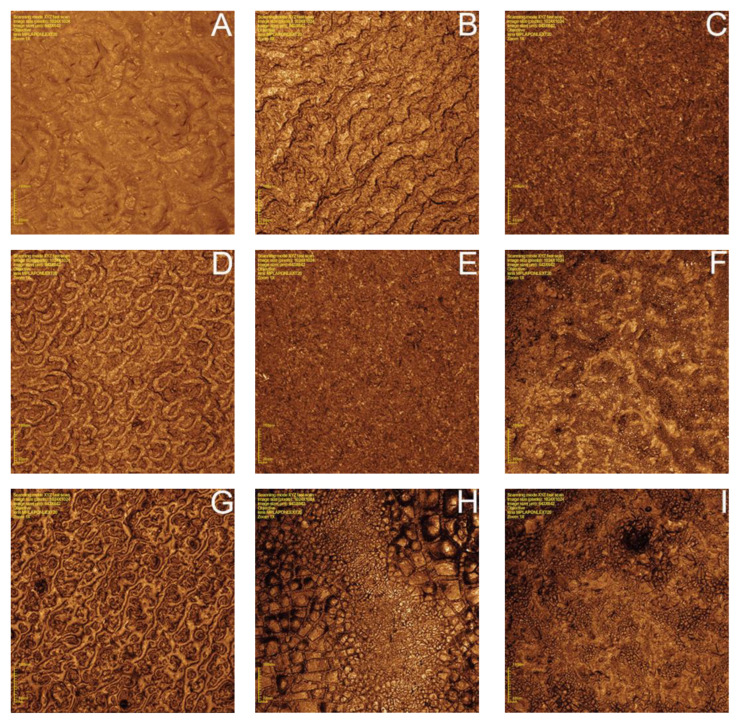
Images of the surface morphology of the disks in different experimental conditions (432× magnification): (**A**) before sintering; (**B**) after sintering; (**C**) silicatization; (**D**) HF 40%; (**E**) HF 40% + silicatization; (**F**) ZrO_2_ nanoparticles; (**G**) SiO_2_ nanoparticles; (**H**) ZrO_2_ + SiO_2_ nanoparticles (50%–50%); (**I**): ZrO_2_ + SiO_2_ nanoparticles (50%–50%) + silane.

**Figure 5 materials-15-07905-f005:**
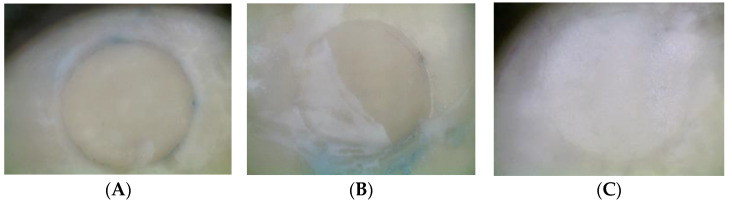
Representative images of failure patterns after the shear test: (**A**) unclassified failure; (**B**) mixed failure; (**C**) adhesive failure.

**Figure 6 materials-15-07905-f006:**
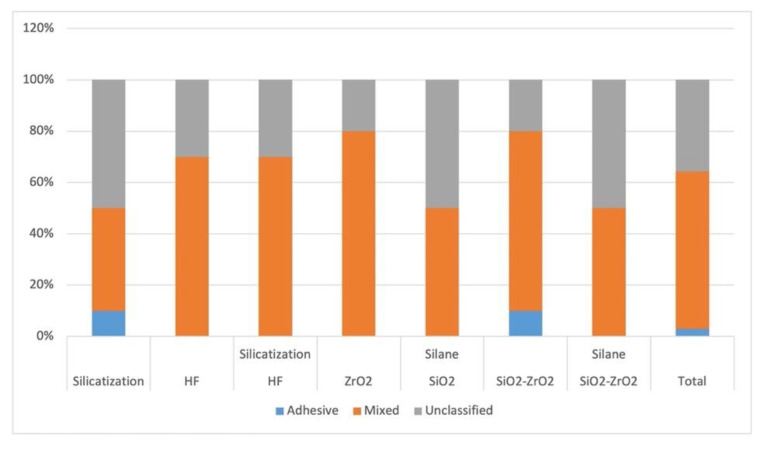
Failure mode analysis.

**Table 1 materials-15-07905-t001:** The contact angles (θa) and surface free energy (γ_s_), and γ_s =_ γ_S_^p^ (polar component) + γ_S_^d^ (dispersive component) values after the different surface treatments.

Groups	Contact Angle θa (°)	Surface Free Energy γ_S_ (mJ.m^−2^)	Polar Component γ_S_^p^ (mJ.m^−2^)	Dispersive Component γ_S_^d^ (mJ.m^−2^)
	Mean (SD)	Mean (SD)	Mean (SD)	Mean (SD)
Control	<3.0	71.89 (0.32) A	45.66 (4.61) A	26.23 (4.88) A
HF	56.0 (12.0) A	52.66 (6.83) B	20.71 (5.46) B	31.95 (2.1) B
HF/ Silicatization	<3.0	71.89 (0.31) A	45.66 (4.61) A	26.23 (4.88) A
ZrO_2_	28.5 (5.0) B	68.05 (0.7) BC	37.86 (0.36) BC	29.81 (0.37) AB
SiO_2_/Silane	13.0 (2.0) B	69.68 (0.2) AC	42.76 (1.07) AC	26.91 (1.03) A
SiO_2_+ZrO_2_	25.3 (3.0) AB	63.77 (5.33) BC	35.67 (2.84) BC	31.39 (4.58) B

Different uppercase letters (in the column) represent significant differences (*p*-value < 0.05).

**Table 2 materials-15-07905-t002:** Surface roughness (Sa) of the Y-TZP discs before sintering, after sintering, and after surface treatments. Results are presented as mean (standard deviation).

Groups	Presintering	After Sintering	After Treatment
	Mean (SD)	Mean (SD)	Mean (SD)
Control	1.72 (0.37) Aa	2.40 (0.32) Ab	2.09 (0.12) Ab
HF	1.71 (0.39) Aa	2.70 (0.49) Ab	2.18 (0.19) Ab
HF/silicatization	1.76 (0.55) Aa	2.59 (0.60) Ab	1.81 (0.10) Ac
ZrO_2_	2.57 (0.49) Aa	3.29 (0.22) Ab	5.20 (0.27) Bc
SiO_2_/Silane	2.08 (0.61) Aa	3.25 (0.77) Ab	7.08 (0.85) Cc
SiO_2_+ZrO_2_	2.10 (0.51) Aa	3.08 (0.59) Ab	4.29 (0.90) Dc
SiO_2_+ZrO_2_/Silane	2.06 (0.60) Aa	2.97 (0.30) Ab	6.19 (0.69) Ec

Different uppercase (in the column) and lowercase (line) letters represent significant differences (*p*-value < 0.05).

**Table 3 materials-15-07905-t003:** Shear bond strength (MPa) after the different surface treatments. Data are presented as mean (standard deviation). (α = 5%).

Groups	Mean (SD)
Control	13.49 (3.80) A
HF	7.62 (2.74) BC
HF/silicatization	9.14 (2.41) AB
ZrO_2_	4.25 (0.3) C
SiO_2_/Silane	10.36 (3.55) AB
SiO_2_+ZrO_2_	7.21 (2.62) BC
SiO_2_+ZrO_2_/Silane	10.44 (4.96) AB

Different uppercase letters (in the column) represent significant differences (*p*-value < 0.05).

**Table 4 materials-15-07905-t004:** Dynamic modulus (GPa) of bar-shaped specimens are presented as mean (standard deviation) before sintering, after sintering, and after surface treatments.

Groups	Dynamic Modulus (GPa)		
	Presintering	After Sintering	After Treatment
	Mean (SD)	Mean (SD)	Mean (SD)
Control	16.40 (0.27) Aa	208.07 (3.90) Ab	205.26 (3.56) Ab
HF	16.62 (0.47) Aa	210.45 (7.24) Ab	213.10 (8.41) Ab
HF/ Silicatization	16.62 (0.66) Aa	207.29 (11.16) Ab	209.36 (3.85) Ab
ZrO_2_	16.98 (0.35) Aa	216.85 (1.82) Ab	211.67 (3.38) Ab
SiO_2_/Silane	16.99 (0.25) Aa	213.69 (3.55) Ab	206.13 (7.12) Ab
SiO_2_+ZrO_2_	17.20 (0.22) Aa	212.78 (3.55) Ab	207.96 (4.16) Ab
SiO_2_+ZrO_2_/Silane	16.74 (0.40) Aa	214.26 (1.47) Ab	206.98 (0.97) Ab

Different uppercase (in the column) and lowercase letters (line) represent significant differences (*p*-value < 0.05).

## Data Availability

Data supporting the results of this study are available in the article and can be requested from the corresponding author.

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
