# Peer review of "Y-TZP Physicochemical Properties Conditioned with ZrO2 and SiO2 Nanofilms and Bond Strength to Dual Resin Cement"

_materials, 2022, doi:10.3390/ma15227905_

Round 1

Reviewer 1 Report

In the review of research article titled: Y-TZP physico-chemical properties conditioned with ZrO2 and SiO2 nanofilms and bond strength to dual resin cement, the authors have described the research work very well covering a lot of aspects. I would like to see this article publish but after some minor modifications as follow;

1.      In the abstract portion, there exist several un-necessary discussion, I would suggest the authors to prepare the concise abstract highlighting the main aspects (values) of their study in abstract.

2.      Authors have not described the drawbacks and missing points in the previous studies which have compelled them to carry out their research work?

3.      Please make the indexing of bands and bonding in the FTIR Figure 3 to make the result more interesting.

4.      In the XRD analysis I would like the authors to make the indexing of the peaks following the description of PDF card number from the literature.

5.      Elemental distribution study of the samples will appeal the readers. I suggest the authors to provide the colored SEM mapping of the samples.

6.      Conclusion is so much descriptive, description should be in results and discussion. Please make the conclusion portion concise.

Author Response

REVIEWER 1

In the review of research article titled: Y-TZP physico-chemical properties conditioned with ZrO2 and SiO2 nanofilms and bond strength to dual resin cement, the authors have described the research work very well covering a lot of aspects. I would like to see this article publish but after some minor modifications as follow;

  1. In the abstract portion, there exist several un-necessary discussion, I would suggest the authors to prepare the concise abstract highlighting the main aspects (values) of their study in abstract.

The Abstract has been corrected according to the reviewer's suggestions. Thank you for the directions.

  1. Authors have not described the drawbacks and missing points in the previous studies which have compelled them to carry out their research work?

Some references were added and the Introduction section was revised to show some results of previous studies that proposed surface treatments to improve shear bond strength. A paragraph was added in the Introduction section to justify the present study once any previous study presented substantial improvement of shear bond strength after surface treatments tested.

  1. Please make the indexing of bands and bonding in the FTIR Figure 3 to make the result more interesting.

Figure 3 was corrected according to reviewer suggestion.

  1. In the XRD analysis I would like the authors to make the indexing of the peaks following the description of PDF card number from the literature.

The XRD analysis was completed according to the reviewer suggestion.

  1. Elemental distribution study of the samples will appeal the readers. I suggest the authors to provide the colored SEM mapping of the samples.

The scanning electron microscope from the Department of Chemistry that we use is out of order, awaiting a technical visit from an international technician, which, at the moment, makes it impossible to obtain the colored SEM mapping of the samples.

  1. Conclusion is so much descriptive, description should be in results and discussion. Please make the conclusion portion concise.

We tried to keep the conclusion concise, but informing the promising results obtained and the condition that generated the best results.

Reviewer 2 Report

The mentioned manuscript is subject to corrections.

1. The Abstract needs to improve. The Abstract mainly contains an enumeration of methods, but there is no general information on the results achieved. The abstract should summarize the findings of the work.

2. Literature review needs to include several recent, relevant publications (high impact) highlighting their key findings. The current version only discussed general aspects while the review of each from several papers is necessary. You may provide a review summary table consisting of a column for the comments or key conclusions.

3. Enhance the objective and novelty of the work in the introduction section.

4. Section 1. To improve the manuscript quality for zirconia, Y-doped zirconia or related materials study, cite the followings: DOIs: 10.1007/s42247-021-00230-5; 10.1021/acsaelm.1c00703.

5. Correlate different sections' results with proper discussion.

6. Quality of all figures is too low. Improve the quality and also increase the font size to make it more visible.

7. The conclusion is too short. In the Conclusion section, state the most important outcome of your work. Do not simply summarize the points already made in the body — instead, interpret your findings at a higher level of abstraction. Show whether, or to what extent, you have succeeded in addressing the need stated in the Introduction.

Author Response

REVIEWER 2

The mentioned manuscript is subject to corrections.

  1. The Abstract needs to improve. The Abstract mainly contains an enumeration of methods, but there is no general information on the results achieved. The abstract should summarize the findings of the work.

The Abstract has been corrected according to the reviewer's suggestions. Thank you for the directions.

  1. Literature review needs to include several recent, relevant publications (high impact) highlighting their key findings. The current version only discussed general aspects while the review of each from several papers is necessary. You may provide a review summary table consisting of a column for the comments or key conclusions.

 Recent publications were added in the Introduction section showing that any substantial improvement of bond strength was achieved with surface treatments already proposed. A review summary table was not added because these tables are more common in review articles.

  1. Enhance the objective and novelty of the work in the introduction section.

A paragraph was added in the Introduction section to show that treatments that have been proposed in the literature did not present substantial improvement of bond strength. Then, surface treatments with nanofilms combined or not with other treatments are proposed to alter surface features of Y-TZP, trying to improve bond strength.

  1. Section 1. To improve the manuscript quality for zirconia, Y-doped zirconia or related materials study, cite the followings: DOIs: 10.1007/s42247-021-00230-5; 10.1021/acsaelm.1c00703.

The manuscripts suggested were read, but the authors understand that they are not directly correlated with what is proposed in this study. We appreciate the indication, but the articles were not cited in this manuscript.

  1. Correlate different sections' results with proper discussion.

 The results were discussed in the Discussion section in the sequence that they were presented in the Results section.

  1. Quality of all figures is too low. Improve the quality and also increase the font size to make it more visible.

High resolution figures have already been submitted separately from the template, as requested in the submission. The font size used was chosen following instructions of the journal, also as in the template. However, the font size was increased to better visualization.

  1. The conclusion is too short. In the Conclusion section, state the most important outcome of your work. Do not simply summarize the points already made in the body — instead, interpret your findings at a higher level of abstraction. Show whether, or to what extent, you have succeeded in addressing the need stated in the Introduction.

Unlike what was asked here, reviewer 1 requested that the Conclusion be more concise, not descriptive. Thus, considering that in the Discussion we sought to address what reviewer 2 requested, we intend to keep the Conclusion straightforward.

Round 2

Reviewer 2 Report

Accept